# C-Flat Turbo: A Faster Path to Continual Learning

## Abstract

Continual Learning (CL) aims to train neural networks on a dynamic task stream without forgetting previously learned knowledge. With the rise of pre-training techniques, strong model generalization has become essential for stable learning. C-Flat is a powerful and general CL training regime that promotes generalization by seeking flatter optima across sequential tasks. However, it requires three additional gradient computations per step, resulting in up to 4× computational overhead. In this work, we propose C-Flat Turbo, a faster yet stronger optimizer with a relaxed scheduler, to substantially reduce training cost. We disclose that gradients toward first-order flatness contain direction-invariant components with respect to the proxy model at $\theta + \epsilon_1^*$, which allows us to skip redundant gradient computations in the perturbed ascent steps. Furthermore, a stage-wise step scheduler and adaptive triggering of the regularization mechanism enable dynamic control of C-Flat behavior throughout training. Experiments demonstrate that our optimizer accelerates most CL methods by at least 1× (up to 1.25×) over C-Flat, while achieving comparable performance. Code will be released upon acceptance.

## 1 Introduction

In the open world, learning systems are requested to absorb new knowledge incrementally, a process known as continual learning (CL) Zhou et al. (2023b); Wang et al. (2023a); Masana et al. (2022). A major obstacle to CL is catastrophic forgetting. To tackle this issue, various approaches have been proposed Zhou et al. (2024a); Wang et al. (2022b); Smith et al. (2023), including memory-based, regularization-based, expansion-based, pre-trained model (PTM)-based methods, etc Wang et al. (2023a); Hadsell et al. (2020); Wang et al. (2023a). Among them, PTM-based methods Jung et al. (2023); Tang et al. (2023); Khan et al. (2023) have shown a more positive effect in overcoming forgetting due to their superior generalization capabilities.

Apart from the methods mentioned above, generalization in the CL optimization process Jia et al. (2022); Tang et al. (2023); Khan et al. (2023); Khattak et al. (2023); Khan et al. (2023) has gradually attracted the attention of the CL community. A series of works demonstrated that sharpness-aware minimization is a powerful training mechanism for maintaining generalization in CL He et al. (2019); Foret et al. (2020); Zhong et al. (2022); Zhuang et al. (2022). These methods focus on flat minima of the CL process, thereby effectively reducing forgetting. For example, an investigation Mehta et al. (2023) proves that sharpness optimization can help overcome forgetting in CL. Recently, C-Flat Bian et al. (2024) is proposed as a promising CL-friendly optimizer to strengthen learning stability on joint knowledge space of new and old tasks Deng et al. (2021); Shi et al. (2021). C-Flat seeks minima lying in a flat neighborhood, which are more likely to be global optima, thereby mitigating forgetting caused by the knowledge gap between different tasks Shi et al. (2021); Kong et al. (2023); Zhuang et al. (2022).

However, flat minima alignment requires calculating a gradient perturbation, which significantly increases computational complexity and overhead He et al. (2019); Foret et al. (2020). For instance, (i) the zeroth-order sharpness Foret et al. (2020) perturbation of C-Flat approximated by the scaled empirical loss requires another backward propagation, which doubles the computational overhead Zhong et al. (2022); Zhang et al. (2023b); Liu et al. (2022). (ii) the first-order flatness in C-Flat, which encourages training to converge to global flat minima Bian et al. (2024), requires calculating a Hessian matrix-related perturbation according to Gradient Norm Aware Minimization (GAM) Zhang et al. (2023b). While this computation can be simplified using the Hessian-vector product or further zeroth-order-based approximations, it still requires two more backpropagation Zhuang et al. (2022).

Moreover, the two-step gradient alignment of C-Flat may enlarge the minima search range, adding extra burden to the fine-tuning phase when learning is nearly converged Hadsell et al. (2020); Bian et al. (2024).

Fortunately, C-Flat features a robust convergence that can converge within far fewer iterations or epochs than SGD Bian et al. (2024). This ensures that C-Flat potentially finishes the CL process faster than SGD Bian et al. (2024) (L3 level of C-Flat). But essentially, the C-Flat optimizer is still inherently plagued by computational overhead. Thus, a natural question arises,

*(Q) Could we find a faster optimization path that guarantees the strong performance of C-Flat while featuring significantly lower computational overhead?*

Through a series of observations and empirical analyses, we find that the gradients of first-order flatness, when projected orthogonally to the gradients of a proxy model at $\rho + \epsilon_1^*$, exhibit more stable changes than even the gradients associated with zeroth-order sharpness during optimization. These insights potentially enable faster paths to continual learning. Therefore, in this work, we decompose the C-Flat gradients into components parallel and orthogonal to the empirical gradient direction, and periodically take shortcuts along the orthogonal directions to more efficiently explore flatter regions of the loss landscape.

Moreover, we observe that the sharpness and flatness gradients not only decrease along with the training loss within each task, but also gradually diminish across tasks, indicating a trend toward increased stability. Based on this, we introduce a stage-wise turbo step scheduler that flexibly adjusts the shortcut frequency. In addition, an adaptive policy is proposed to combine C-Flat with SGD for further efficiency improvements.

Our technical contributions are three-fold:

(i) We identify the invariant flat-driven branches in the first-order regime and propose an efficient variant, C-Flat Turbo, which selectively shortcuts along stable directions toward flatter regions.

(ii) We uncover the stabilization of sharpness-aware gradients in continual learning and propose a stage-wise linear scheduler combined with an adaptive triggering mechanism to dynamically regulate the behavior of C-Flat during training.

(iii) Extensive experiments show that C-Flat Turbo outperforms state-of-the-art methods and achieves more than 1× speedup (up to 1.25×), while maintaining performance.

## 2 RELATED WORK

**Continual learning.** Along this line, continual learning can be taxonomized into three groups Wang et al. (2023a); Hadsell et al. (2020) as follows, *(i) Memory-based methods* base a limited budget in a memory to resist forgetting Rebuffi et al. (2017); Rolnick et al. (2019); Jeeveswaran et al. (2023); Sun et al. (2023b). Many efforts selectively store a few representative exemplars for rehearsal during CL. Apart from direct replay, some other efforts Deng et al. (2021); Lin et al. (2022); Saha et al. (2020); Lin et al. (2023); Sun et al. (2023a) resort to proxy of former knowledge as memory to overcome forgetting, e.g., bases of gradient space, representative prototype, etc. *(ii) Regularization-based methods* are characterized by introducing favorable regularization terms to trade off new and old knowledge Li & Hoiem (2018); Kirkpatrick et al. (2017); Cha et al. (2021). Trivially, consensus practices include weight Rudner et al. (2022); Kim et al. (2022); Akyürek et al. (2022), function Li & Hoiem (2018); Oh et al. (2022), or feature regularization Bhat et al. (2023); Gao et al. (2022), which encourage these spaces to remain close to their original states. Like natural cognitive systems, this consolidation helps to limit the renewal of important terms. *(iii) Expansion-based methods* aim to dynamically modularize network structures towards each task to tackle forgetting Zhou et al. (2023c). Methods in this group construct task-specific parameters or architecture (e.g., parameter allocation Liu et al. (2021); Abati et al. (2020), model decomposition/mixture, and modular network Yan et al. (2021); Zhu et al. (2022)) to explicitly reduce inter-task interferenceSerrà et al. (2018); Hu et al. (2023); Yoon et al. (2020). In other words, these efforts shift the burden of storing raw data to the retention architecture Zhou et al. (2023c).

**Continual learning using generalization.** The strong generalizability of PTMs further advances the CL Zhou et al. (2024a). Adhere to Zhou et al. (2024a), we taxonomize these studies into three

groups as follows, *(i) Prompt-based methods* leverage prompt learning to enable lightweight updates to PTM Jia et al. (2022). Many efforts Wang et al. (2022b); Smith et al. (2023); Jung et al. (2023); Tang et al. (2023); Khan et al. (2023) resort to various prompt selections, e.g., the way of prompt retrieval, task-special prompts, and prompt generation. Moreover, some work extends the concept of prompts to broader scenes Gan et al. (2023); Khattak et al. (2023); Khan et al. (2023), i.e., visual prompts, pre-trained vision-language model prompts, etc. Overall, these efforts strike a balance between the generalizability of PTM and the encoding of information from downstream tasks. *(ii) Representation-based methods* focus on building classifiers by leveraging the generalization capability of PTM Zhou et al. (2023d). Briefly, this line of work mainly dedicates the calibration of classifiers and optimize their internal relationships Zhou et al. (2024b); McDonnell et al. (2024); Zhang et al. (2023a); Zhu et al. (2021). Like, concatenating the backbone to improve classifier representation, using random projection to remove class-wise correlations, and replaying features to calibrate the classifier. *(iii) Model mixture-based methods.* introduce a set of models and utilize various model mixture approaches (model ensemble, model merge, etc.) to generate final outputs Wang et al. (2023b); Gao et al. (2023); Zhou et al. (2023e); Chen et al. (2023); Wang et al. (2022a). Trivially, the method in this category integrates several PTM with strong generalization capabilities to produce more robust results Zhou et al. (2024a). Yet, some of the drawbacks that arise are associated with these efforts, e.g. extra computational overhead and memory buffer.

**Better and efficient generalization in continual learning.** In CL, research into how flat minima He et al. (2019); Foret et al. (2020); Baldassi et al. (2020) affect catastrophic forgetting is still at a nascent stage Chaudhry et al. (2018); Lopez-Paz & Ranzato (2017). A few works are done to examine flatness evaluation during CL in some special instances Deng et al. (2021); Shi et al. (2021), e.g., tuning parameters within the flatness minima regions. Promisingly, C-Flat Bian et al. (2024), a CL-tailored optimizer that first introduces the concept of continual flatness to overcome forgetting, which lifts off a new line for CL. The above efforts scrutinize the mechanism of flat minima in CL, yet they suffer from reduced optimization efficiency due to entrapment in many random perturbations Liu et al. (2022); Du et al.; 2021), especially involving successive tasks. In this paper, we advance an efficient version of C-Flat to boost CL.

## 3 METHOD

### 3.1 RETHINKING THE MECHANISM OF C-FLAT

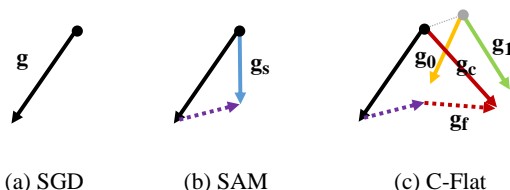

(a) SGD      (b) SAM      (c) C-Flat

Figure 1: Brief illustration of C-Flat Bian et al. (2024). (a) SGD optimizes along the negative direction of the gradients, $g = \nabla L(f(\theta^T))$. (b) SAM Foret et al. (2020) computes the gradients $g_s$ at an adversarially perturbed position $\theta + \rho \cdot g/\|g\|$, and then updates the original model parameters. (c) C-Flat Bian et al. (2024) further calculates the first-order flatness gradient $g_f$, based on the perturbed parameters $\theta + \rho \cdot (g_s - g)/\|(g_s - g)\|$.

Let $\mathcal{S}^t = \{(\mathbf{x}_i^t, \mathbf{y}_i^t)\}_{i=1}^{n^t}$ denote the training set with $n^t$ samples for task $t$, and let $\ell(f(\mathbf{x}; \boldsymbol{\theta}), \mathbf{y})$ be the per-sample loss of the neural network $f$ with parameters $\boldsymbol{\theta}$ on a data point $(\mathbf{x}, \mathbf{y})$. Continual Learning (CL) aims to learn a model $f$ with parameters $\boldsymbol{\theta} \in \mathbb{R}^d$ that minimizes the statistical risk across all tasks seen up to the current task $T$, under the constraint of limited or no access to previous data $\mathcal{S}^t$ for $t < T$. Specifically, CL methods optimize the model parameters $\boldsymbol{\theta}_T$ during training on task $T$ by minimizing the empirical loss over the available data: $\boldsymbol{\theta}_T = \arg\min_{\boldsymbol{\theta}} \mathcal{L}(\boldsymbol{\theta}, \mathcal{S}^T)$, where $\mathcal{L}(\boldsymbol{\theta}, \mathcal{S}^T) = \frac{1}{n^T} \sum_{i=1}^{n^T} \ell(f(\mathbf{x}_i^T; \boldsymbol{\theta}), \mathbf{y}_i^T)$. Depending on the specific CL methods, the training data may include a mixture of current task data $\mathcal{S}^T$ and additional data, such as stored exemplars or reconstructed samples from previous tasks. For simplicity, we omit $\mathcal{S}^T$ and denote $\boldsymbol{\theta}$ as $\boldsymbol{\theta}^T$ later.

As shown in Figure 1, recent C-Flat methods Bian et al. (2024) mitigate catastrophic forgetting by jointly optimizing for zeroth-order sharpness $\mathcal{R}_\rho^0(\boldsymbol{\theta})$ and first-order flatness $\mathcal{R}_\rho^1(\boldsymbol{\theta})$, encouraging the model to find a parameter region with uniformly low loss and curvatures. The optimization objective on task $T$ can be formulated as:

$$\min_{\boldsymbol{\theta}} \ \mathcal{L}(\boldsymbol{\theta}) + \mathcal{R}_\rho^0(\boldsymbol{\theta}) + \lambda \cdot \mathcal{R}_\rho^1(\boldsymbol{\theta}), \tag{1}$$

where $\lambda > 0$ is a balancing hyperparameter, and $\rho > 0$ defines the neighborhood radius around the current parameter $\theta$. Specifically, the zeroth-order term is defined as:

$$\mathcal{R}_\rho^0(\theta) = \max_{\|\epsilon_0\| \le \rho} \mathcal{L}(\theta + \epsilon_0) - \mathcal{L}(\theta) \tag{2}$$

which seeks uniformly low loss within a local neighborhood of $\theta$. The first-order flatness term is:

$$\mathcal{R}_\rho^1(\theta) = \rho \cdot \max_{\|\epsilon_1\| \le \rho} \left\|\nabla\mathcal{L}(\theta + \epsilon_1)\right\|, \tag{3}$$

which encourages low curvature of the loss landscape. Here, $\|\cdot\|$ denotes the $\ell_2$ norm.

**Propagation flow.** Assuming that the loss function $\mathcal{L}$ is differentiable and bounded, and following the derivation of $\nabla R_\rho^0(\theta)$ in Foret et al. (2020); Singh & Alistarh (2020); Zhang et al. (2023b), we can approximate $\nabla R_\rho^1(\theta)$ as follows. More notations are provided in Appendix A.2.

$$\nabla\mathcal{R}_\rho^0(\theta) = \nabla\mathcal{L}(\theta + \epsilon_0^*) - \nabla\mathcal{L}(\theta), \quad \epsilon_0^* \approx \rho \cdot \frac{\nabla\mathcal{L}(\theta)}{\|\nabla\mathcal{L}(\theta)\|}. \tag{4}$$

$$\nabla\mathcal{R}_\rho^1(\theta) \approx \rho \cdot \nabla\left\|\nabla\mathcal{L}(\theta + \epsilon_1^*)\right\| \approx \nabla\mathcal{L}\left(\theta + \epsilon_1^* + \rho \cdot \frac{\nabla\mathcal{L}(\theta + \epsilon_1^*)}{\|\nabla\mathcal{L}(\theta + \epsilon_1^*)\|}\right) - \nabla\mathcal{L}\left(\theta + \epsilon_1^*\right),$$

$$\epsilon_1^* \approx \rho \cdot (g_s - g)/(\|g_s - g\|). \tag{5}$$

**Notations.** Here, we define several variations and terminology that will frequently be used later. Detailed descriptions can be found in Appendix A.1.

i) **the empirical loss term:** $g = \nabla\mathcal{L}(\theta)$ as the original gradients derived from the vanilla optimizer.

ii) **the zeroth-order sharpness term:** $g_s = \nabla\mathcal{L}(\theta + \epsilon_0^*)$ as the perturbed SAM gradients, where $g_s - g = \nabla\mathcal{R}_\rho^0(\theta)$ measures the sharpness of the loss landscape.[1]

iii) **the first-order flatness term:** $g_f = \nabla\mathcal{R}_\rho^1(\theta) = g_1 - g_0$ as the regularization gradients, which quantifies the flatness of the loss landscape. Here, the intermediate gradients are given by $g_0 = \nabla\mathcal{L}(\theta + \epsilon_1^*)$ and $g_1 = \nabla\mathcal{L}\left(\theta + \epsilon_1^* + \rho \cdot \frac{\nabla\mathcal{L}(\theta + \epsilon_1^*)}{\|\nabla\mathcal{L}(\theta + \epsilon_1^*)\|}\right)$.

As shown above, optimizing the gradients of the C-Flat objective can be divided into three components: *the empirical loss term $g$, the sharpness term $g_s - g$, and the flatness term $g_f$*. Among them, $g$ is required in every update step to reduce the empirical risk, whereas $g_s$ and $g_f$ incur one and two additional backward passes, respectively, using perturbed models $\theta^T + \epsilon_0^*$ and $\theta^T + \epsilon_1^*$. Both regularization terms contribute to identifying flatter regions. Despite the performance benefits, searching for such regions is computationally expensive and can impose a heavy burden on CL systems. Therefore, extracting time-efficient solutions is imperative.

## 3.2 C-FLAT TURBO

In this section, we introduce C-Flat Turbo, an enhanced variant of C-Flat designed to accelerate training in continual learning scenarios. C-Flat Turbo primarily explores the invariant direction of flatness promotion and skips redundant computations required for first-order flatness gradients. In addition, we propose an adaptive mechanism that monitors sharpness online for selectively using C-Flat with vanilla optimizers. The following experiments mainly start from pretrained models, providing better initialization and reducing the impact of lossy approximations in C-Flat Turbo.

### 3.2.1 TAKING SHORTCUTS TOWARD FLATNESS

Base optimizers like SGD and Adam reduce the loss function along gradient directions. However, the resulting model parameters are often sensitive to small perturbations or noise, which can lead to

---

[1] We use $g_s - g$ rather than $g_{vs}$ to measure sharpness, for better alignment with the flatness term $g_f = g_1 - g_0$ used later. Though it is not fully orthogonal to the gradient direction $g$, it still captures the direction that promotes zeroth-order sharpness, and has been widely used in Zhao et al. (2022a); Zhang et al. (2023b) when combined with the vanilla gradient $g$ to form a sharpness-aware update direction.

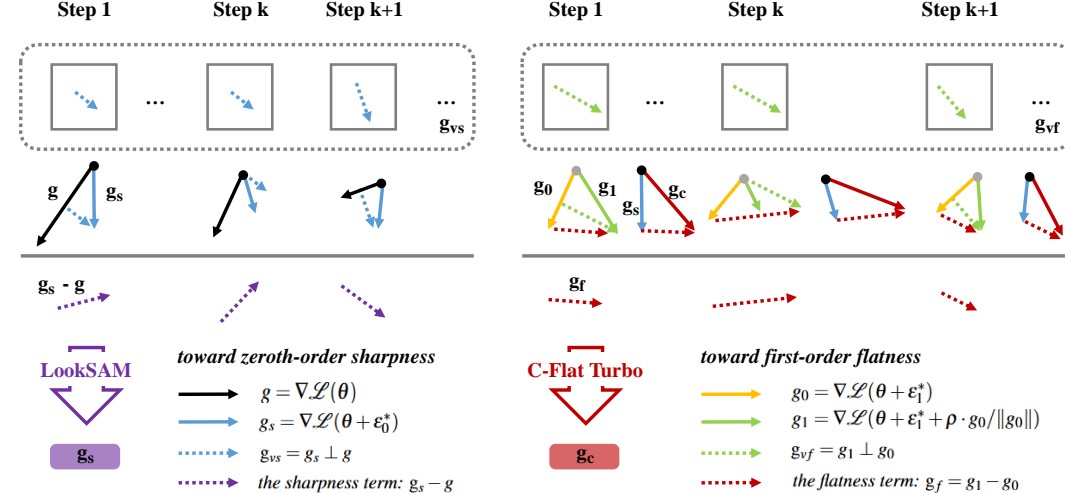

Figure 2: Schematic illustration of C-Flat Turbo. **Left:** LookSAMLiu et al. (2022) decomposes the SAM gradients into two components: one parallel to $g$ that reduces the loss, and an orthogonal component $g_{vs}$ that guides convergence to a common low-loss region. Empirical results show that $g_{vs}$ exhibits significantly slower variation compared to $g$. **Right:** C-Flat Turbo investigates the latent invariance of $g_{vf}$, the flatness component orthogonal to the gradients at the perturbed model $\theta + \epsilon_1^*$, which exhibits even slower variation than $g_{vs}$ in LookSAM.

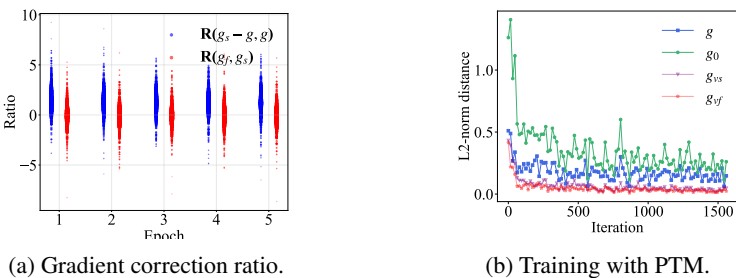

(a) Gradient correction ratio.
(b) Training with PTM.

Figure 3: **Left:** Gradient correction ratio distributions for $g_s - g$ and $g_f$ across training epochs. With more data distributed around the sides, the difference becomes more pronounced. **Right:** We visualize the L2-norm distances between gradients and those from five steps ago during training. Changes in sharpness ($g_{vs}$) and flatness ($g_{vf}$) related directions vary more slowly than those along the base gradient directions ($g$ and $g_0$).

degradation of previously learned parameters when adapting to new tasks. Seeking a unified low-loss landscape within a local region around the global minima has proven effective for continual learning. C-Flat builds upon this idea by promoting flatter regions via first-order flatness, in conjunction with zeroth-order sharpness.

In the context of zeroth-order sharpness $g_s - g$, the component $g_{vs}$, defined as $g_{vs} = g_s \sin(\phi_s)$ where $\phi_s$ denotes the angle between $g$ and $g_s$, is orthogonal to the primary gradient direction $g$. It has been shown to facilitate the exploration of flatter regions while changing more slowly than $g$, thereby guiding the search for low loss regions, as illustrated in Figure 2. This observation further motivates us to investigate whether similar direction invariant components exist in the optimization of the first-order flatness term.

To visualize the optimization dynamics of $g_s - g$ and $g_f$ relative to their reference directions, $g$ for SAM and $g_s$ for C-Flat, we define the gradient correction ratio as $\text{ratio}(m, n) = \log\left(\left|\frac{m}{n+\epsilon}\right|\right)$, $\epsilon = 10^{-12}$, and then conduct experiments in EASE over 5 epochs. Figure 3a illustrates the distributions of $(g_s - g, g)$ and $(g_f, g_s)$ in Task 1. It can be observed that $g_s - g$ exhibits heavier tails than $g_f$ in the early stages, indicating stronger adjustment introduced by zeroth-order sharpness. As the optimization converges to a local minimum, sharpness and flatness collaborate in the search for flatter regions, with sharpness still playing a dominant role. This phenomenon suggests that $g_f$ serves as a smaller rectification upon SAM, even subtler than the correction $g_s - g$ induced by SAM itself.

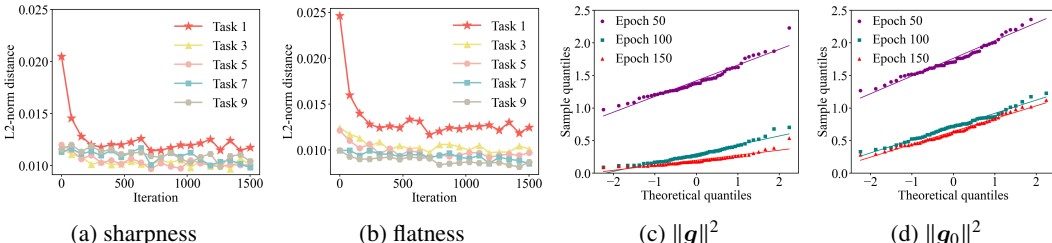

(a) sharpness      (b) flatness      (c) $\|g\|^2$      (d) $\|g_0\|^2$

Figure 4: **Left:** Gradient variation across tasks during CL. Sharpness and flatness vary significantly in the first stage, then stabilize. **Right:** Q-Q plots of $\|g\|^2$ and $\|g_0\|^2$, both gradually approximate a normal distribution during learning.

In approximating $\mathcal{R}_\rho^1(\boldsymbol{\theta})$ in Equation 5, $\boldsymbol{\epsilon}_1^*$ naturally emerges as a small perturbation direction aligned with the sharpness term. Consequently, the point $\boldsymbol{\theta}^T + \boldsymbol{\epsilon}_1^*$ serves as a proxy model for computing the flatness term. As illustrated in Figure 1, $\boldsymbol{g}_f$ fulfills a role similar to SAM, but is defined in the perturbed model $\boldsymbol{\theta}^T + \boldsymbol{\epsilon}_1^*$ located in a region of high curvature. In this sense, $\boldsymbol{g}_f$ embodies an invariant direction toward flatness, denoted as $\boldsymbol{g}_{vf}$, analogous to the gradient $\boldsymbol{g}_0$ in its proxy model. Here, $\boldsymbol{g}_{vf}$ is defined as $\boldsymbol{g}_{vf} = \boldsymbol{g}_1 \sin(\phi_f)$, where $\phi_f$ denotes the angle between $\boldsymbol{g}_0$ and $\boldsymbol{g}_1$.

Figure 3b illustrates the gradient differences across iterations, measured by the L2-norm distance from those in the previous five iterations. As observed, $\boldsymbol{g}_{vs}$ fluctuates more slowly than $\boldsymbol{g}$, consistent with the findings in Liu et al. (2022). Furthermore, $\boldsymbol{g}_0$ appears more unstable than $\boldsymbol{g}$, reflecting its high-curvature nature and sensitivity to parameter changes. Interestingly, despite the considerable variation in $\boldsymbol{g}_0$, the flatness oriented $\boldsymbol{g}_f$ remains substantially more stable, even exceeding the stability of $\boldsymbol{g}_{vs}$, in line with the observations in Figure 3a. These insights support extracting the vertical component from $\boldsymbol{g}_f$ with respect to $\boldsymbol{g}_0$, which serves as a direction promoting flatness, as illustrated in Figure 2. Consequently, for the subsequent $k - 1$ steps, the cached $\boldsymbol{g}_{vf}$ can be reused to efficiently guide the search for low curvature directions based on the proxy model at $\boldsymbol{\theta} + \boldsymbol{\epsilon}_1^*$. In other words, we can add $\boldsymbol{g}_{vf}$ to $\boldsymbol{g}_0$ with a coefficient $\beta$. This strategy avoids recomputing $\boldsymbol{g}_1$ when evaluating flatness in the proxy model, thereby reducing computational cost.

### 3.2.2 Dynamic control for C-Flat

**Stage-wise turbo step scheduling.** As continual learning progresses, the model's parameter space becomes increasingly flatter, with earlier classes becoming more distinctly separated. Figures 4a and 4b depict the evolution of the sharpness and flatness terms over time. Both gradients fluctuate considerably during the initial incremental stage but stabilize in later stages, with this effect being particularly pronounced for flatness. This observation motivates assigning smaller turbo steps to earlier tasks and larger steps to later ones. To this end, we introduce a linear scheduler that adaptively adjusts the step size for computing sharpness and flatness aware gradients throughout training. It is simply formulated as Turbo-$k_0$ with $k_t = k_0 + 10 \cdot n/N$, where $k_0$ and $k_t$ denote the initial and task-$n$ step sizes, respectively, with a total of $N$ tasks. Empirically, C-Flat Turbo equipped with this scheduler achieves substantial speedup while maintaining competitive performance.

**Adaptive triggering of regularization.** Existing research on zeroth-order sharpness has proposed reducing computational overhead by combining SAM updates with standard ERM. For example, SS-SAM Zhao et al. (2022b) employs a Bernoulli trial to determine whether to apply SAM, while AE-SAM Jiang et al. (2023) applies SAM adaptively only when the sharpness measure falls below a dynamically updated threshold. However, first-order flatness has received relatively little attention. Figures 4c and 4d visualize the distributional properties of $\|g\|^2$ and $\|g_0\|^2$ using quantile-quantile (Q-Q) plots. Points that lie closer to the reference line indicate that the corresponding variable is closer to following a normal distribution. Based on this observation, we can use exponential moving average (EMA) updates to estimate the mean and variance of $\|g_0\|^2$, following Jiang et al. (2023):

$$\mu_{f,j} = \delta\mu_{f,j-1} + (1 - \delta)\|g_{0j}\|^2, \qquad \sigma_{f,j}^2 = \delta\sigma_{f,j-1}^2 + (1 - \delta)(\|g_{0j}\|^2 - \mu_{f,j})^2, \qquad (6)$$

where $j$ denotes the current iteration and $\delta = 0.9$ is the decay factor used to discount outdated gradient values. The flatness regularization is triggered only when $\|g_{0j}\|^2 > \mu_{f,j} + \sigma_{f,j}$. A detailed description of the overall optimization procedure is provided in Algorithm 1 in the Appendix.

Table 1: Accuracy and training speeds using five state-of-the-art methods, with a pretrained ViT-B/16-IN1K backbone. Red and green denote the baseline and the efficient optimizer. **Bolded** indicates best performance.

| Model | Method | CIFAR100 B0_Inc10 | | CUB B0_Inc10 | | IN-R B0_Inc20 | | ObjNet B0_Inc10 | | Img/s↑ |
|---|---|---|---|---|---|---|---|---|---|---|
| | | Avg↑ | Last↑ | Avg↑ | Last↑ | Avg↑ | Last↑ | Avg↑ | Last↑ | |
| Typical | iCaRL Rebuffi et al. (2017) | 77.83 | 66.64 | 82.91 | 74.00 | 72.13 | 61.62 | 48.06 | 28.20 | 73.35 (100%) |
| | +*C-Flat Bian et al. (2024)* | 79.72 | 67.15 | 83.47 | 74.81 | 72.92 | 62.35 | 49.59 | 29.03 | 19.72 (26.9%) |
| | +*C-Flat Turbo* | **79.82** | **68.54** | **84.00** | **75.12** | **73.11** | **62.38** | **50.49** | **29.30** | 45.89 (62.6%) |
| | MEMO Zhou et al. (2023c) | 82.26 | 73.89 | 86.66 | 79.90 | 70.96 | 61.05 | 56.22 | 38.32 | 135.14 (100%) |
| | +*C-Flat Bian et al. (2024)* | 82.61 | 75.49 | 87.03 | 80.73 | 71.69 | 62.93 | 56.50 | 39.45 | 46.11 (34.1%) |
| | +*C-Flat Turbo* | **83.02** | **75.76** | **87.15** | **80.92** | **72.38** | **63.55** | **57.52** | **40.62** | 91.91 (68.0%) |
| PTM-based | L2P Wang et al. (2022b) | 89.36 | 83.94 | 73.04 | 59.14 | 78.05 | 72.60 | 64.18 | 52.10 | 110.29 (100%) |
| | +*C-Flat Bian et al. (2024)* | 89.56 | 84.35 | **74.36** | **62.11** | 78.67 | 73.78 | 64.53 | 52.47 | 28.63 (30.0%) |
| | +*C-Flat Turbo* | **89.78** | **84.69** | 74.12 | 61.97 | **78.86** | **73.82** | **64.64** | **52.55** | 65.50 (59.4%) |
| | Ranpac McDonnell et al. (2024) | 94.32 | 90.72 | 92.61 | 88.68 | 82.07 | 76.80 | 71.66 | 60.17 | 154.64 (100%) |
| | +*C-Flat Bian et al. (2024)* | 94.41 | 90.70 | 92.67 | 88.76 | 82.66 | 77.25 | 72.15 | 60.33 | 42.98 (27.8%) |
| | +*C-Flat Turbo* | **94.45** | **90.74** | **93.12** | **89.02** | **83.13** | **77.83** | **72.16** | 60.33 | 94.34 (61.0%) |
| | EASE Zhou et al. (2024b) | 91.91 | 87.30 | 89.16 | 83.96 | 80.49 | 75.05 | 64.38 | 52.02 | 166.67 (100%) |
| | +*C-Flat Bian et al. (2024)* | 92.05 | 87.91 | 89.37 | 84.05 | 80.97 | 75.64 | 64.89 | 52.47 | 44.25 (26.5%) |
| | +*C-Flat Turbo* | **92.36** | **87.96** | **89.56** | **84.18** | **81.18** | **75.76** | **64.96** | **52.61** | 102.74 (61.6%) |

## 4 EXPERIMENTS

### 4.1 EXPERIMENT SETUP

**Datasets.** Following Zhou et al. (2024a), we perform the evaluation on CIFAR100 Krizhevsky et al. (2009), CUB200 Wah et al. (2011), ImageNet-R Hendrycks et al. (2021) (IN-R), and ObjectNet Barbu et al. (2019) (ObjNet). These datasets contain 100 classes in CIFAR100, 200 classes in CUB200, and cover ImageNet-R and ObjectNet, which exhibit large domain gaps relative to the pre-trained datasets (ImageNet Deng et al. (2009)). Following Zhou et al. (2024a; 2023b), we denote the data split as 'B-*m* Inc-*n*', meaning that the initial task contains *m* classes, and each subsequent task contains *n* classes. The random seed for class-order shuffling is fixed at 1993 Zhou et al. (2023a; 2024a).

**Baselines**. Typical CL and pre-trained model (PTM)-based CL methods Zhou et al. (2024a) are used to assess C-Flat Turbo. For the former, we cover the classical iCaRL Rebuffi et al. (2017) and MEMO Zhou et al. (2023c) methods. For the latter, we compare against L2P Wang et al. (2022b), Ranpac McDonnell et al. (2024), and EASE Zhou et al. (2024b), spanning common CL categories.

**Implementation details.** All experiments and compared methods are implemented and reproduced using PyTorch and PILOT Zhou et al. (2024a; 2023a); Bian et al. (2024). If not specialized, all hyperparameters or configurations remain unchanged in the open-source repositoryZhou et al. (2024a). To ensure a fair comparison, we evaluate all methods with the same model and vanilla SGD optimizer (Adam for L2P) and adopt ViT-B/16-IN1K as the representative pre-trained models Wang et al. (2022b); Zhou et al. (2023d). Implementation details can be found in section A.4. For evaluation, we primarily present the results in terms of average accuracy (Avg), final task accuracy (Last), and the average running speed (Img/s) across all tasks.

### 4.2 FASTER AND STRONGER PERFORMANCE

We thoroughly evaluate the performance of C-Flat Turbo. As shown in Table 1, although pre-trained models exhibit strong generalization capabilities, their feature spaces remain susceptible to contamination during continual adaptation to evolving data distributions, thereby exacerbating catastrophic forgetting. While C-Flat mitigates this degradation through flat region search mechanisms, it incurs significant computational overhead. On the one hand, C-Flat Turbo addresses this limitation by freely taking shortcuts toward flatness, extracted from earlier steps, without repeated computation. On the other hand, it flexibly combines C-Flat with base optimizers through a stage-wise step schedule and an adaptive trigger for regularization, further accelerating the training process. A more detailed per-task accuracy progression and ablation studies are provided in the Appendix A.6. Experimental results demonstrate that C-Flat Turbo achieves better accuracy than C-Flat, while significantly reducing training time. Notably, we find that C-Flat Turbo remains stable even in PTM scenarios with larger generalization gaps (e.g., CUB200, ImageNet-R, and ObjectNet). C-Flat Turbo trains CL models at about 2× the speed of C-Flat, and 0.6× that of SGD. Overall, whether applied to typical

Table 2: Accuracy trained from scratch with ResNet-18 and ResNet-34. **Bolded** indicates best result.

| Method | | ResNet-18 | | | ResNet-34 | |
| | Avg↑ | Last↑ | Img/s↑ | Avg↑ | Last↑ | Img/s↑ |
| --- | --- | --- | --- | --- | --- | --- |
| iCaRL Rebuffi et al. (2017) | $59.13_{\pm0.30}$ | $41.23_{\pm0.91}$ | 2333.3(100%) | $58.80_{\pm1.04}$ | $41.26_{\pm0.99}$ | 1250.4(100%) |
| +C-Flat Bian et al. (2024) | $59.45_{\pm0.18}$ | $42.47_{\pm0.06}$ | 686.3(29.4%) | $59.55_{\pm0.93}$ | $42.09_{\pm0.61}$ | 359.8(28.8%) |
| +C-Flat Turbo | $\mathbf{59.84_{\pm0.05}}$ | $\mathbf{42.84_{\pm0.18}}$ | 1750.1(75.0%) | $\mathbf{59.75_{\pm0.55}}$ | $\mathbf{42.34_{\pm0.54}}$ | 960.6(76.8%) |
| MEMO Zhou et al. (2023c) | $48.63_{\pm0.78}$ | $29.19_{\pm0.89}$ | 2413.8(100%) | $68.49_{\pm1.74}$ | $57.05_{\pm1.46}$ | 1873.2(100%) |
| +C-Flat Bian et al. (2024) | $49.98_{\pm0.61}$ | $30.76_{\pm0.57}$ | 886.1(36.7%) | $69.00_{\pm1.39}$ | $59.29_{\pm0.73}$ | 569.1(30.4%) |
| +C-Flat Turbo | $\mathbf{50.51_{\pm0.55}}$ | $\mathbf{32.24_{\pm0.27}}$ | 1891.9(78.4%) | $\mathbf{69.48_{\pm1.25}}$ | $\mathbf{59.33_{\pm0.65}}$ | 1372.5(73.3%) |

CL benchmarks or to scenarios with large domain gaps in PTM, C-Flat Turbo maintains strong performance while offering superior training efficiency compared to the baseline C-Flat optimizer.

## 4.3 TRAINING FROM SCRATCH

Table 2 presents the accuracy results of iCaRL and MEMO, using ResNet-18 and ResNet-34 trained without pre-trained models. Notably, iCaRL benefits significantly from C-Flat Turbo, achieving an increase of 1.61% in last accuracy on ResNet-18 and 1.08% on ResNet-34. Similarly, MEMO exhibits substantial gains, particularly in final stage accuracy, where C-Flat Turbo improves performance by 3.05% on ResNet-18 and 2.28% on ResNet-34. Besides, it is noted that C-Flat Turbo forgets less than C-Flat across training due to its soft constraint on sharpness around local minima. The consistent improvements across different backbone architectures highlight the robustness of C-Flat Turbo in mitigating catastrophic forgetting while maintaining computational efficiency. Furthermore, the larger performance gains observed in MEMO suggest that C-Flat Turbo is particularly beneficial for expansion-based approaches, which involve multiple module updates and often struggle with stability and adaptability. These findings validate C-Flat Turbo as a highly effective strategy for continual learning, offering substantial accuracy improvements while enabling flexible training strategies.

## 4.4 COMPARISON TO OTHER OPTIMIZERS

In Table 3, we report the average accuracy, last accuracy, and training speeds on the CIFAR100 B0_Inc10 setting compared to various zeroth-order optimizers. In special, the step of all sharpness in LookSAM and flatness in C-Flat Turbo is fixed to 5.

For the performance, as shown in Table 3, we first observe that SAM and LookSAM do not offer obvious benefits over the vanilla optimizer, but C-Flat series shows significant improvement. The reason is that the parameters of backbones loaded from the pre-trained model already possess strong generalization, resulting in uniformly low losses around local minima across various parameter perturbations. Nevertheless, C-Flat Turbo further surgeries the strong generalization of the pre-trained model by the horizontal and vertical components of the oracle gradient, thus boost CL.

For the training speeds, as concluded in Table 3, although LookSAM significantly accelerates training compared to SAM by reusing historical gradients, it degrades performance on EASE due to its single zeroth-order regularization and the simplistic use of past iteration gradients. C-Flat Turbo differs from LookSAM in that it progressively updates sharpness gradients to vanilla components and imposes more strict constraints to encourage convergence to a flatter region. Compared to C-Flat, our efficient method achieves a speedup of approximately 1x (from 30.0% to 59.4% and from 26.5% to 61.6%) on L2P and EASE, which is even faster than the SAM.

## 4.5 ABLATION STUDIES

**Parameter Sensitivity.** In Figure 5a, we empirically investigate the sensitivity of the scale factor $\beta$ and the sampling step $k$ in EASE, trained under the CIFAR-100 B0_Inc10 setting. Here, $k$ denotes the frequency of leveraging the sharpness and flatness properties in C-Flat Turbo. For fairness, the sharpness and flatness steps are kept equal, meaning that C-Flat Turbo-$k$ indicates the cached sharpness and flatness gradients are updated every $k$ iterations. The hyperparameters $\rho$ and $\lambda$ are fixed at 0.05 and 0.2, respectively. Figure 5a demonstrates that $\beta = 0.8$ is the optimal choice in most cases. Moreover, when $k = 2$ and $k = 5$, the performance exhibits similar accuracy fluctuations as long as $\beta$ lies within the range [0.4, 1.0].

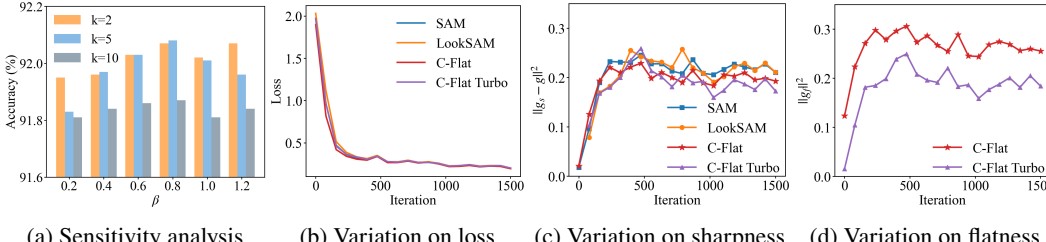

| (a) Sensitivity analysis | (b) Variation on loss | (c) Variation on sharpness | (d) Variation on flatness |

Figure 5: Evolution of sharpness and flatness on EASE w/ and w/o C-Flat Turbo.

Table 3: Accuracy and training speeds of L2P and EASE across various optimizers. Task orders are shuffled for evaluation on dynamic sequences. Red and green denote the baseline and the efficient optimizer. **Bolded** indicates best performance.

| Method | L2P Wang et al. (2022b) | | | EASE Zhou et al. (2024b) | | |
|---|---|---|---|---|---|---|
| | Avg↑ | Last↑ | Img/s↑ | Avg↑ | Last↑ | Img/s↑ |
| Vanilla | $87.92_{\pm1.30}$ | $83.66_{\pm1.49}$ | 110.29 (100%) | $91.16_{\pm0.71}$ | $87.49_{\pm0.32}$ | 166.67(100%) |
| *+SAM Foret et al. (2020)* | $88.18_{\pm1.39}$ | $83.84_{\pm1.39}$ | 56.60 (51.3%) | $91.36_{\pm0.82}$ | $87.61_{\pm0.27}$ | 86.71(52.0%) |
| *+LookSAM Liu et al. (2022)* | $88.35_{\pm1.39}$ | $83.80_{\pm1.20}$ | 88.76 (80.5%) | $90.89_{\pm0.86}$ | $86.99_{\pm0.28}$ | 132.74(79.6%) |
| *+C-Flat Bian et al. (2024)* | $88.62_{\pm0.88}$ | $84.04_{\pm0.60}$ | 28.63 (30.0%) | $91.60_{\pm1.18}$ | $87.69_{\pm0.50}$ | 44.25 (26.5%) |
| *+C-Flat Turbo* | $\mathbf{89.57_{\pm0.36}}$ | $\mathbf{84.48_{\pm0.09}}$ | 65.50 (59.4%) | $\mathbf{91.75_{\pm0.80}}$ | $\mathbf{87.74_{\pm0.20}}$ | 102.74 (61.6%) |

**Evolution of Sharpness and Flatness.** We approximate the sharpness and flatness gradients using cached branches and current SGD directions, following the same optimization procedure as C-Flat. This efficient approach does not slow convergence. As shown in Figure 5b, C-Flat Turbo converges as fast as other optimizers. Figure 5c illustrates that all optimizers start with low sharpness initially, owing to the pretrained backbone's generalization, and then sharpness declines alongside the loss $\mathcal{L}(\boldsymbol{\theta}_\rho)$. Notably, the sharpness and flatness gradients in C-Flat Turbo converge to lower values than in C-Flat, due to the intermediate gradient descent steps being free of regularization constraints.

## 4.6 DISCUSSION ON SCHEDULER CHOICES

Figure 6 compares the accuracy and training speed of MEMO and EASE. For the linear scheduler, we increase the step size $k$ with the task number $n$, formulated as $k = 5 + 10 \cdot n/N$, whereas for the version without a scheduler, we keep $k$ fixed at 5. MEMO expands its architecture for each new task, increasing both parameters and computational cost, which prolongs training. As shown in Figure 6, C-Flat Turbo with or without the scheduler, consistently outperforms vanilla C-Flat in accuracy. In terms of speed, both variants are faster, with the scheduler achieving approximately 15% additional speedup. In contrast, EASE reuses frozen adapters, keeping training costs stable. Figure 6 demonstrates that the scheduler provides roughly 30% speedup over C-Flat as $k$ increases, while maintaining comparable accuracy.

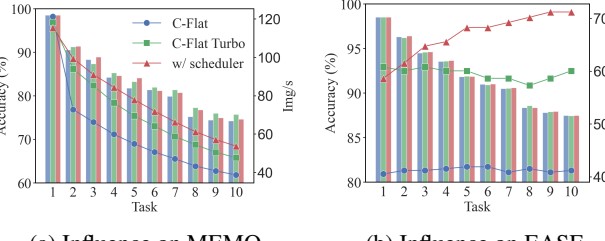

| (a) Influence on MEMO | (b) Influence on EASE |

Figure 6: Accuracy and training speed of scheduling or not.

## 5 CONCLUSION

This paper proposes C-Flat Turbo, which selectively takes shortcuts along stable directions toward flatness for more efficient optimization. Building upon the observation that flatness gradients diminish across tasks, we additionally introduce a linear scheduler that adaptively adjusts the turbo steps, as well as an adaptive trigger for selectively applying C-Flat regularization. In summary, this work reveals that adaptively modifying the oracle gradient can yield tangible efficiency gains during continual learning. Moreover, C-Flat Turbo can be seamlessly integrated into any CL method, offering at least a 1× speedup.

ETHICS STATEMENT

Our work introduces C-Flat Turbo, a faster yet stronger optimizer with a relaxed scheduler, to substantially reduce training cost. We do not foresee any direct ethical risks associated with this work, as it focuses purely on optimization techniques rather than data generation or manipulation. Nevertheless, we firmly state that C-Flat Turbo is currently a research-oriented project.

REPRODUCIBILITY STATEMENT

We are committed to ensuring the reproducibility of our work. All essential details for reproducing our results are provided within the paper and the appendix. The details of architectures and their training details are presented in section 4 and section A. Our technical contribution, C-Flat Turbo, are described with comprehensive details in section 3. The evaluation protocols are detailed in section 4.1. We will ensure the release of our source code, pre-trained model weights, and evaluation scripts upon publication of this work. We can also provide any source code if any reviewer asks for a detailed implementation.

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
