# A APPENDIX

## A.1 SYMBOLS AND NOTATIONS

- model parameter: $\boldsymbol{\theta}$;
- SAM perturbed model parameter: $\boldsymbol{\theta} + \boldsymbol{\epsilon}_0^* = \boldsymbol{\theta} + \rho \cdot \frac{\nabla \mathcal{L}(\boldsymbol{\theta})}{\|\nabla \mathcal{L}(\boldsymbol{\theta})\|}$;
- proxy model parameter: $\boldsymbol{\theta} + \boldsymbol{\epsilon}_1^* = \boldsymbol{\theta} + \rho \cdot (\boldsymbol{g}_s - \boldsymbol{g})/\|\boldsymbol{g}_s - \boldsymbol{g}\|$;
- proxy perturbed model parameter: $\boldsymbol{\theta} + \boldsymbol{\epsilon}_1^* + \rho \cdot \nabla \mathcal{L}(\boldsymbol{\theta} + \boldsymbol{\epsilon}_1^*)/\|\nabla \mathcal{L}(\boldsymbol{\theta} + \boldsymbol{\epsilon}_1^*)\|$;
- the empirical loss: $\mathcal{L}(\boldsymbol{\theta})$, with its gradient $\boldsymbol{g}$;
- the SAM loss: $\mathcal{L}(\boldsymbol{\theta}) + \mathcal{R}_\rho^0(\boldsymbol{\theta}) = \max_{\|\boldsymbol{\epsilon}_0\| \leq \rho} \mathcal{L}(\boldsymbol{\theta} + \boldsymbol{\epsilon}_0)$ with its gradient $\boldsymbol{g}_s$;
- the C-Flat loss: $\mathcal{L}(\boldsymbol{\theta}) + \mathcal{R}_\rho^0(\boldsymbol{\theta}) + \lambda \cdot \mathcal{R}_\rho^1(\boldsymbol{\theta}) = \max_{\|\boldsymbol{\epsilon}_0\| \leq \rho} \mathcal{L}(\boldsymbol{\theta} + \boldsymbol{\epsilon}_0) + \rho \cdot \max_{\|\boldsymbol{\epsilon}_1\| \leq \rho} \left\|\nabla \mathcal{L}(\boldsymbol{\theta} + \boldsymbol{\epsilon}_1)\right\|$, with its gradient $\boldsymbol{g}_s + \boldsymbol{g}_f$;
- the gradient of proxy model: $\boldsymbol{g}_0 = \nabla \mathcal{L}(\boldsymbol{\theta} + \boldsymbol{\epsilon}_1^*)$;
- the gradient of proxy perturbed model: $\boldsymbol{g}_1 = \nabla \mathcal{L}(\boldsymbol{\theta} + \boldsymbol{\epsilon}_1^* + \rho \cdot \nabla \mathcal{L}(\boldsymbol{\theta} + \boldsymbol{\epsilon}_1^*)/\|\nabla \mathcal{L}(\boldsymbol{\theta} + \boldsymbol{\epsilon}_1^*)\|)$;
- the empirical loss term: $\boldsymbol{g} = \nabla \mathcal{L}(\boldsymbol{\theta})$;
- the zeroth-order sharpness term: $\boldsymbol{g}_s - \boldsymbol{g} = \nabla \mathcal{R}_\rho^0(\boldsymbol{\theta})$;
- the first-order flatness term: $\boldsymbol{g}_f = \nabla \mathcal{R}_\rho^1(\boldsymbol{\theta})$;

## A.2 DERIVATION OF EQUATION 5

Following Bian et al. (2024); Zhang et al. (2023b), the gradient of the first-order flatness loss $\mathcal{R}_\rho^1$ is:

$$\nabla_{\boldsymbol{\theta}} \mathcal{R}_\rho^1(\boldsymbol{\theta}) = \rho \cdot \nabla_{\boldsymbol{\theta}} \max_{\boldsymbol{\epsilon} \in B(0,\rho)} \|\nabla \mathcal{L}(\boldsymbol{\theta} + \boldsymbol{\epsilon})\| \tag{7}$$

$$= \rho \cdot \nabla_{\boldsymbol{\theta}} \|\nabla \mathcal{L}(\boldsymbol{\theta} + \boldsymbol{\epsilon}_1^*)\|$$

$$= \rho \cdot \left( \frac{\partial}{\partial \boldsymbol{\theta}} \|\nabla \mathcal{L}(\boldsymbol{\theta} + \boldsymbol{\epsilon}_1^*)\| + \frac{\partial \boldsymbol{\epsilon}_1^*}{\partial \boldsymbol{\theta}} \cdot \nabla_{\boldsymbol{\epsilon}} \|\nabla \mathcal{L}(\boldsymbol{\theta} + \boldsymbol{\epsilon})\| \big|_{\boldsymbol{\epsilon} = \boldsymbol{\epsilon}_1^*} \right)$$

$$\approx \rho \cdot \nabla_{\boldsymbol{\theta}} \|\nabla \mathcal{L}(\boldsymbol{\theta} + \boldsymbol{\epsilon}_1^*)\| \quad \text{// neglect higher-order term for tractability}$$

Here, $\boldsymbol{\epsilon}_1^*$ denotes the optimal perturbation that maximizes the gradient norm within the $\ell_2$-ball $B(0, \rho)$. To make the computation tractable, we approximate it using first-order Taylor expansion and finite differences:

$$\boldsymbol{\epsilon}_1^* = \arg \max_{\boldsymbol{\epsilon} \in B(0,\rho)} \|\nabla \mathcal{L}(\boldsymbol{\theta} + \boldsymbol{\epsilon})\| \tag{8}$$

$$\approx \arg \max_{\boldsymbol{\epsilon} \in B(0,\rho)} \left( (\nabla_{\boldsymbol{\theta}} \|\nabla \mathcal{L}(\boldsymbol{\theta})\|)^T \boldsymbol{\epsilon} \right) \quad \text{// first-order Taylor expansion}$$

$$= \rho \cdot \frac{\nabla_{\boldsymbol{\theta}} \|\nabla \mathcal{L}(\boldsymbol{\theta})\|}{\|\nabla_{\boldsymbol{\theta}} \|\nabla \mathcal{L}(\boldsymbol{\theta})\|\|} \quad \text{// direction of steepest ascent}$$

$$\approx \rho \cdot \frac{\nabla \mathcal{L}(\boldsymbol{\theta} + \boldsymbol{\delta}) - \nabla \mathcal{L}(\boldsymbol{\theta})}{\|\nabla \mathcal{L}(\boldsymbol{\theta} + \boldsymbol{\delta}) - \nabla \mathcal{L}(\boldsymbol{\theta})\|} \quad \text{// finite-difference approximation}$$

$$= \rho \cdot \frac{\boldsymbol{g}_s - \boldsymbol{g}}{\|\boldsymbol{g}_s - \boldsymbol{g}\|},$$

where $\boldsymbol{g} = \nabla \mathcal{L}(\boldsymbol{\theta})$, $\boldsymbol{g}_s = \nabla \mathcal{L}(\boldsymbol{\theta} + \boldsymbol{\delta})$, and $\boldsymbol{\delta} = \rho' \cdot \frac{\boldsymbol{g}}{\|\boldsymbol{g}\|}$ is a small perturbation in the direction of the gradient, with $\rho' \ll \rho$.

Let $\boldsymbol{\theta}_p = \boldsymbol{\theta} + \boldsymbol{\epsilon}_1^*$ be the perturbed model after maximizing the gradient norm. Then Equation 7 continues as:

$$\nabla_{\boldsymbol{\theta}} \mathcal{R}_\rho^1(\boldsymbol{\theta}) \approx \rho \cdot \nabla_{\boldsymbol{\theta}} \left\|\nabla \mathcal{L}(\boldsymbol{\theta}_p)\right\| \tag{9}$$

$$\approx \frac{\rho}{\rho'} \cdot \left[ \nabla \mathcal{L}\left(\boldsymbol{\theta}_p + \rho' \cdot \frac{\nabla \mathcal{L}(\boldsymbol{\theta}_p)}{\|\nabla \mathcal{L}(\boldsymbol{\theta}_p)\|}\right) - \nabla \mathcal{L}(\boldsymbol{\theta}_p) \right],$$

where $\rho' \ll \rho$ is a small step size for finite-difference approximation.

Then, the direction-invariant component of $g_f$ with respect to $g_0$ is defined as:

$$g_{vf} = g_f - \frac{\langle g_f, g_0 \rangle}{\|g_0\|^2} \cdot g_0, \tag{10}$$

which represents the orthogonal projection of $g_f$ onto the subspace perpendicular to $g_0$ (i.e., $g_{vf} \perp g_0$), capturing a direction-invariant update toward flatness.

### A.3 C-Flat Turbo Algorithm

---

**Algorithm 1** C-Flat Turbo

---

**Input:** Training phase $T$, training data $S^T$, model parameter $\theta$, total iterations $J$, oracle loss function $\mathcal{L}$, learning rate $\eta$, C-Flat coefficient $\lambda$, Turbo step $k$, $\mu_{s,0} = \mu_{f,0} = 0$, $\sigma_{s,0} = \sigma_{f,0} = 1e-8$.

**Output:** Model trained at the current time $T$ with C-Flat.

1: **for** $j = 1$ to $J$, sample batch $B_j^T$ from dataset $S^T$ **do**
2:      Compute gradient: $g = \nabla \mathcal{L}(\theta)$
3:      Initialize update direction: $\bar{g} = g$
4:      Update EMA statistics: $\mu_{s,j}, \sigma_{s,j}, \mu_{f,j}, \sigma_{f,j}$ by eq. (6)
5:      **if** $\|g\|^2 \geq \mu_{s,j} + \sigma_{s,j}$ **then**
6:          **if** $j \bmod k = 0$ **then**
7:              Compute $g_s = \nabla \mathcal{L}(\theta + \epsilon_0^*) - \nabla \mathcal{L}(\theta)$ by eq. (4)
8:              Cache $g_{vs} = g_s \sin(\phi_s)$, where $\phi_s$ denotes the angle between $g$ and $g_s$
9:          **else**
10:              Simulate $g_s = g + \beta \frac{\|g\|}{\|g_{vs}\|} g_{vs}$
11:          **end if**
12:          Update direction: $\bar{g} = g_s$
13:      **end if**
14:      **if** $\|g_0\|^2 \geq \mu_{f,j} + \sigma_{f,j}$ **then**
15:          **if** $j \bmod k = 0$ **then**
16:              Compute $g_f = \rho \cdot \nabla \left\| \nabla \mathcal{L}(\theta + \epsilon_1^*) \right\|$ using $g_0$ and $g_1$ by eq. (5)
17:              Cache $g_{vf} = g_f \sin(\phi_f)$, where $\phi_f$ denotes the angle between $g_0$ and $g_1$
18:          **else**
19:              Simulate $g_f = g_0 + \beta \frac{\|g_0\|}{\|g_{vf}\|} g_{vf}$
20:          **end if**
21:          Update direction: $\bar{g} = \bar{g} + \lambda \cdot g_f$
22:      **end if**
23:      Update turbo step $k$ according to Section 4.6
24:      Update model parameter: $\theta^T = \theta^T - \eta \cdot \bar{g}$
25: **end for**

---

### A.4 Hyperparameter Configurations

Here, we primarily provide the hyperparameter configurations for methods trained on CIFAR100. For other datasets, we follow the original settings from the open repository and keep $k = 5$ and $\beta = 0.8$ fixed.

### A.5 Memory Usage

The cached gradients are used to substitute partial sharpness-aware gradient computations, so their memory usage heavily depends on the number of trainable parameters in the model. As shown in the table 5, although larger models require more cached gradients, the overall memory overhead remains almost negligible relative to the expansion typically caused by large architectures.

Table 4: Hyperparameter settings for CIFAR100.

|        | Epochs | LR    | BS | Tasks | Exemplar/class | $\rho$ | $\lambda$ | $k$ | $\beta$ |
|--------|--------|-------|----|-------|----------------|--------|-----------|-----|---------|
| iCaRL  | 20     | 1e-3  | 32 | 10    | 20             | 0.1    | 0.2       | 5   | 0.8     |
| MEMO   | 20     | 1e-3  | 32 | 10    | 20             | 0.1    | 0.2       | 5   | 0.8     |
| L2P    | 5      | 2e-3  | 16 | 10    | -              | 0.02   | 0.2       | 5   | 0.8     |
| Ranpac | 5      | 1e-2  | 16 | 10    | -              | 0.05   | 0.2       | 5   | 0.8     |
| EASE   | 5      | 2.5e-3| 16 | 10    | -              | 0.05   | 0.2       | 5   | 0.8     |

Table 5: Memory usage of different architecture.

| Method | Optimizer | Backbone | Training backbone | Trainable / total params | Memory |
|--------|-----------|----------|-------------------|--------------------------|--------|
| EASE   | C-Flat    | ViT-Base-16 | ×              | 1.19M / 86.99M           | 2.14GB |
|        | Turbo     |          |                   |                          | 2.15GB |
|        | C-Flat    | ViT-Large-16 | ×             | 3.17M / 306.47M          | 5.34GB |
|        | Turbo     |          |                   |                          | 5.37GB |
| iCaRL  | C-Flat    | ResNet-18 | ✓               | 11.17M / 11.17M          | 1.55GB |
|        | Turbo     |          |                   |                          | 1.66GB |
|        | C-Flat    | ResNet-34 | ✓               | 21.28M / 21.28M          | 2.32GB |
|        | Turbo     |          |                   |                          | 2.51GB |

## A.6 Per-task accuracy and ablation studies

Per-task accuracy provides a more detailed view of the continual learning process. As shown in Table 6, the reuse mechanism significantly reduces training speed with minimal performance loss, while the linear scheduler for step size further enhances speed, particularly for longer tasks. The adaptive trigger additionally accelerates training, as it allows basic single propagation gradient descent in certain stages. Regarding performance gains, prior works have shown that selectively applying SAM updates can outperform applying SAM throughout training. For instance, SS-SAM explicitly demonstrates that with appropriate scheduling, models can achieve comparable or even superior performance at substantially lower computational cost compared to training exclusively with SAM. Similar observations also have been reported for AE-SAM and SAM-In-Later-Phase.

Table 6: Per-task accuracy and ablation study results for EASE trained on the 10-split CIFAR100 dataset.

| Method | reuse | sche. | trigger | T1 | T2 | T3 | T4 | T5 | T6 | T7 | T8 | T9 | T10 | Avg | Img/s |
|--------|-------|-------|---------|------|------|------|------|------|------|------|------|------|------|------|--------|
| EASE   | ×     | ×     | ×       | 98.40 | 96.25 | 94.63 | 93.88 | 91.80 | 90.92 | 90.47 | 88.09 | 87.58 | 87.17 | 91.92 | 166.67 |
| +C-Flat| ×     | ×     | ×       | 98.50 | 96.45 | 94.87 | 94.08 | 91.94 | 91.05 | 90.64 | 88.44 | 87.93 | 87.58 | 92.15 | 44.25 |
| +Turbo | ✓     | ×     | ×       | 98.50 | 96.37 | 94.77 | 93.94 | 91.92 | 91.05 | 90.67 | 88.28 | 87.81 | 87.45 | 92.08 | 67.20 |
|        | ✓     | ✓     | ×       | 98.40 | 96.31 | 94.74 | 93.89 | 91.90 | 91.00 | 90.70 | 88.25 | 87.75 | 87.40 | 92.03 | 74.63 |
|        | ✓     | ✓     | ✓       | 98.50 | **96.60** | **95.07** | **94.15** | **92.08** | **91.27** | **90.73** | **88.52** | **88.00** | 87.57 | **92.25** | **102.74** |

## A.7 Detail Distance Evolution of Gradients

Figure 7 shows the L2-norm distances between sharpness and flatness gradients and their reference gradients across tasks. While $g$ and $g_0$ exhibit significant fluctuations during training, the gradients $g_{vs}$ and $g_{vf}$, core to zeroth-order sharpness and first-order flatness regularization, change much more slowly. This stability suggests their potential as shortcut directions for flat region exploration, bypassing the need for model ascent and backpropagation.

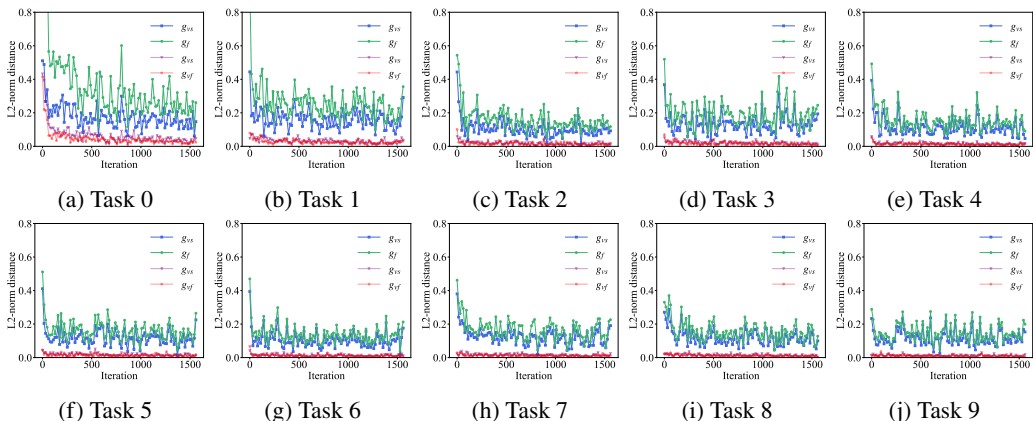

Figure 7: Visualization of L2-norm distances of the gradients every 5 steps across 10 tasks.

### A.8 LIMITATIONS AND FUTURE WORK

This work is based on the stabilization evolution of sharpness and flatness during C-Flat optimization, where we approximate regularization terms using progressively memorized branches. Although efficient, there is still room for improvement in comparison to the speed of vanilla optimizers, particularly in minimizing the computational overhead of $g_f$ in flatness. Then, the current framework has focused on validation using PTM-based and typical CIL tasks, but its application to other CL tasks, such as Vision-Language Models (VLMs), remains unexplored. Extending our methodology to these diverse CL settings could reveal its broader applicability and provide valuable insights into its generalization capabilities. Future work could also explore the integration of C-Flat Turbo with advanced learning paradigms, such as few-shot learning and lifelong learning, to evaluate its potential in real-world, dynamic environments.