# OpenReview forum: "C-Flat Turbo: A Faster Path to Continual Learning"
_ICLR.cc/2026/Conference — ICLR 2026 Conference Withdrawn Submission_

### Official Review · Reviewer_1zUD · 2025-10-26

**Soundness:** 3
**Presentation:** 2
**Contribution:** 2
**Rating:** 4
**Confidence:** 3

**Summary:**

Motivated by the observed stabilization of sharpness-aware gradients in continual learning, this paper introduces C-Flat Turbo, a variant that selectively shortcuts along stable directions toward flatter regions. It further proposes a stage-wise linear scheduler, coupled with an adaptive triggering mechanism, to dynamically regulate C-Flat’s behavior during training. And compared with the C-Flat method, the approach delivers better performance and faster speed.

**Strengths:**

1. This paper’s motivation is clear, and it offers a concrete critique of C-Flat’s limitations, most notably the heavy computational overhead of the first-order flatness term, which requires computing g and g1 at a proxy point (i.e., two extra backpropagations per step).
2. The writing is fluent and the structure is sound. The paper presents its motivation, method, and results in a clear, logical order, with figures and notation that generally support comprehension. Section transitions are smooth, and the key assumptions and design choices are stated explicitly.

**Weaknesses:**

1. Limited practical impact. Reported speedups are only relative to C-Flat; the method remains slower than original training, while accuracy gains are very small (often <0.5%). The paper lacks a compelling compute–accuracy trade-off to justify adoption in realistic CL settings.
2. Narrow evidence and weak theoretical grounding. The claim that gf is a subtler correction than gs, g is drawn mainly from Fig. 3 (EASE, ~5 epochs). There is no systematic validation of a theoretical account why this should hold.
3. Incomplete compute reporting. Experiments emphasize throughput/wall-clock but omit hardware-agnostic metrics , peak memory, and energy (e.g., J/sample). Sensitivity to k,β,λ,ρ and the resulting complexity Pareto curves are also missing.

**Questions:**

1. From the experimental results, the claimed speedup is only relative to C-Flat. It is still slower than the vanilla methods, and the accuracy gains are very marginal. Thus, I question the practical significance of this improvement over C-Flat.It is necessary to strengthen the explanation of whether such plug-in methods are worth using in continual learning.

2. The proposed method appears to rely solely on the phenomenon visualized in Figure 3, obtained from experiments in EASE over just 5 epochs, to infer that (g_f) serves as a smaller rectification upon SAM, even subtler than the correction (g_s - g) induced by SAM itself. Is this observation accidental? It seems to lack evidence of generality or any supporting theoretical justification.

3. On the experimental side, there are the following issues:
a) Across the four datasets, the improvements are generally within 0.5%. Is this because these datasets are not suitable to showcase the method, or is the method’s effect inherently limited?
b)The abstract states that C-Flat results in up to 4× computational overhead, yet this paper only reports speed comparisons. It lacks comparisons of computational resources (e.g., FLOPs, forward/backward counts, peak memory, energy).

---

### Official Review · Reviewer_8v6u · 2025-10-30

**Soundness:** 3
**Presentation:** 2
**Contribution:** 2
**Rating:** 4
**Confidence:** 3

**Summary:**

The paper presents a sharpness-aware optimization method for continual learning, extending the C-Flat framework. The authors propose alternative strategies to explore flatter regions of the loss landscape while significantly reducing the computational overhead introduced by C-Flat’s additional gradient computations.
After exploring both zeroth-order sharpness and first-order flatness, the authors identify an invariant direction that effectively promotes flatness.  They leverage this directional invariance to enable the reuse of previously computed components through a caching mechanism and an adaptive triggering strategy.
The proposed method is evaluated against both pretrained and non-pretrained baselines, as well as existing sharpness-aware optimizers. Experimental results show that C-Flat Turbo outperforms prior methods, achieving comparable or superior continual learning performance while substantially lowering computational cost compared to the original C-Flat.

**Strengths:**

- The paper introduces the core concepts in a logically coherent sequence, effectively linking empirical findings on loss landscape dynamics to the formal derivation of the proposed regularization and optimization mechanisms
- The results appear to substantiate the main hypothesis, demonstrating consistent improvements in stability and supporting the intuition behind the proposed method. Moreover, the proposed C-Flat Turbo significantly reduces computational cost compared to C-Flat while maintaining (or improving) performance.

**Weaknesses:**

- While the paper provides a clear rationale for leveraging invariant directions in the optimization of flatness, there remains an asymmetry between how sharpness and flatness are treated. Although the intuition is understandable, this design choice would benefit from a clearer justification, as the underlying reasoning for not reusing $g_{vs}$​ remains implicit (see question).
- Since the paper is presented as an improvement over C-Flat and builds upon the stabilization of sharpness and flatness during C-Flat optimization, one would expect a more direct comparison with the baselines used in the original C-Flat paper (e.g., Replay, WA, PODNet, DER, FOSTER, etc.).  However, these baselines are missing, and since the experimental setup also changes (e.g., number of tasks, class-per-task scheduling, or initial task size). This makes it difficult to assess the consistency and fairness of the reported gains.
- Although the proposed approach clearly reduces the overhead of C-Flat, its overall computational cost still appears substantial. It is therefore not entirely clear whether the observed gains justify the use of C-Flat (and its Turbo variant) over standard optimizers in continual learning settings.

Minor
- The current use of in-text citations makes the paper difficult to read, especially in the introduction and related work sections. Please use parentheses consistently and ensure correct usage of `\cite`, `\citet`, and `\citep`.
- There are a few potential typos: for instance, line 308 should be _Turbo-kt​_, and line 373 should read _scheduler_ instead of _schedule_. Also lines 206-207 seem to be out of context.
- The notation for speedup is somewhat confusing. Typically, “1×” denotes no improvement (same speed). Consider rephrasing using “2×” (for double speed) or stating explicitly “100–125% improvement” to avoid ambiguity.
- Some details appear to be missing or unclear. For example, in Table 2 it is not specified which dataset is used for training. Please verify that all experimental setups are fully described. The metrics are not clearly presented. Since continual learning metrics are not universally standardized, consider adding a brief description or a supplementary section defining the metrics used (especially “Avg” and “Last”). The nomenclature can be confusing without explicit clarification.

**Questions:**

1. The paper mentions that both the sharpness and flatness components exhibit slowly varying, direction-invariant behavior. However, only the flatness component is reused via caching in C-Flat Turbo. Could you elaborate on why this asymmetry exists? Why is the caching mechanism applied only to the flatness component, even though both sharpness and flatness are described as direction-invariant?
2. Could the authors clarify or quantify how many gradient computations are skipped at most when both the _scheduler_ and the _adaptive trigger_ are active?  In other words, what is the maximum reduction in backward passes per iteration that the method achieves under the full optimization scheme?
3. How does C-Flat Turbo perform on the baselines reported in the original C-Flat paper but omitted here (e.g., Replay, WA, PODNet, DER, FOSTER)?  If these experiments were not possible, how do the authors justify their absence?
4. The same clarification would be useful for the changes in experimental settings (e.g., number of tasks, class-per-task schedule).
5. How were the confidence intervals in Tables 2 and 3 computed? Would it be possible to report them also for Table 1?
6. Were any statistical significance tests conducted to validate the reported improvements?
7. Could the authors formally define the metrics used for evaluation?

---

### Official Review · Reviewer_vfyP · 2025-11-01

**Soundness:** 2
**Presentation:** 1
**Contribution:** 2
**Rating:** 4
**Confidence:** 4

**Summary:**

The paper proposes C-Flat Turbo as a faster and more effective version of  C-Flat for CL problems. Instead of recomputing the first-order flatness term from scratch at each step, the main idea is to decompose it into a "vertical" component $g_{vf} = g_1 sin(\theta_f)$ with respect to $g_0$, defined as the gradient with respect to the proxy model $\theta+\epsilon_1$. By caching $g_{vf}$ for the next steps, it can be added to $g_0$ using a coefficient $\beta$, hence avoiding the recomputation of the whole $g_1$. On top of that, the authors proposed both a stage-wise scheduler for the step size and an adaptive trigger, which uses EMAs to trigger the sharpness regularization only when needed. Complete experiments have been performed, showing the practical speed and efficiency of the proposed method with respect to C-flat and other sharpness-aware optimizers.

**Strengths:**

- The positioning of the paper is very interesting, as it describes a method for CL that takes into consideration the sharpness/curvature of the loss landscape. This research direction is extremely relevant as it can be impactful for the efficiency of real-world pretrained models.
- The idea of the paper is simple and straightforward, enabling its use with a wide array of other CL proposals
- The experiments are complete and show the very strong empirical capabilities of C-Flat Turbo, which is able to improve over C-Flat both the training speed and the final accuracy.

**Weaknesses:**

- The strongest weakness, in my opinion, is the presentation of the paper. I arrived at the end of the C-Flat Turbo section without having a clear idea of what C-Flat Turbo does. Actually, it seems that the only place where it is explicitly defined what C-Flat Turbo is on the description of Figure 2. I also found the notation to be quite confusing and sometimes not well-defined (I will provide examples in the Questions) and many concepts are used before any actual definition or explanation, making hard to follow the paper.

- While the empirical results of the paper are very strong, they are the only real contribution of this work. In my opinion, the complete lack of theoretical results and explanations strongly weakens the proposal. Moreover, the proposed C-Flat Turbo, appears to be only a small tweak around the already published C-Flat, which is a smart tweak, but scientifically not very impactful. Finally, as the only strong contribution is the empirical results, it seems strange to me not to share the code.

**Questions:**

Here are some questions and notes that I hope the authors can answer/improve:

- There are some small writing issues. Some examples: "C-Flat features a robust convergence that can converge" (I do not think that the convergence can converge), "More notations are provided in Appendix..." (I think that the notation should be univocal on the whole paper), "the empirical loss term : g=..." (I do not understand why a gradient is callled empirical loss term). I hope the paper can be revised to be more clear.

- In the context of the paper I do not think it is true that "...Adam reduces the loss function along gradient directions" as in the original paper of Adam, it is well explained that the curvature information is considered through an approximation of the Fisher information. What do you mean?

- In line 264 a ratio is defined which is never used again in the main text. In the following lines some parenthesis are used, but without recalling the ratio function.

- In line 258 the authors affiirm that $g_vs$ facilitates the exploration of flatter regions, but no reference or theoretical results are provided. What am I missing?

- i would really appreciate a deeper explanation of the main concept of the paper which is briefly presented in line 286 "$g_f$ embodies an invariant direction toward flatness...". Why? Invariant to what?

- I would like a definition of "vertical component" as I am not sure of the meaning of "vertical" in this context.

- in section 3.2.2 the "turbo steps" term is used without having defined previously what these turbo steps are. What are they?

- Finally, many of the results of the paper are based on some results in Figure 2 and 3. I personally believe that this is a weak way of presenting the results, and I would appreciate a deeper explanation of Figure 2 (which I think it is not clear). A paper fully based on a couple of observation about some figures can be very weak. If possible, more theoretical results on the main paper would be greatly appreciated.

I appreciate the work of the authors, but I think it is still not complete. My final rating can easily change (increasing or decreasing) based on the authors answers.

---

### Official Review · Reviewer_ryUf · 2025-11-01

**Soundness:** 2
**Presentation:** 2
**Contribution:** 2
**Rating:** 2
**Confidence:** 5

**Summary:**

C-Flat Turbo is a practical advancement in continual-learning optimization that achieves ~2× speedup by caching the slowly-evolving orthogonal component \(g_{vf}\) of the flatness gradient, eliminating redundant backward passes while preserving flat-minima properties.

The method uses stage-wise scheduling and adaptive triggering to balance efficiency with stability, representing a data-driven refinement that integrates curvature-aware theory with engineering pragmatism for scalable continual learning.

However, the theoretical foundation remains incomplete: \(g_{vf}\) reuse is empirically motivated without formal convergence proofs or approximation-error bounds, leaving open questions about how gradient drift affects optimization trajectories.

In summary, C-Flat Turbo is a promising, empirically validated tool that demonstrates gradient-reuse can maintain stability and generalization, though future work connecting these heuristics to rigorous optimization theory would strengthen its theoretical standing.

**Strengths:**

1. Practical Efficiency Improvement
C-Flat Turbo achieves roughly \(2\times\) speedup over the original C-Flat while maintaining comparable or slightly better accuracy across diverse CL benchmarks (CIFAR-100, CUB-200, IN-R, ObjectNet).
The reuse of the first-order flatness gradient’s orthogonal component \(g_{vf}\) effectively removes redundant backward passes, reducing training overhead without altering the optimization target.

2. Empirical Robustness and Plug-and-Play Design
Demonstrates consistent improvement on both from-scratch (ResNet-18/34) and PTM-based (ViT-B/16) continual learning setups.
Can be seamlessly integrated into existing methods (iCaRL, MEMO, L2P, Ranpac, EASE) via a simple optimizer-level replacement, confirming its general plug-in compatibility.

3. Dynamic Adaptation Mechanisms
Introduces a stage-wise scheduler \(k_t = k_0 + 10\cdot n/N\) to reduce flatness updates as tasks stabilize, and an adaptive trigger based on the EMA of \(\|g_0\|^2\) to activate regularization only when curvature increases.
These heuristics yield further efficiency gains while preventing over-regularization.

4. Empirical Observation-Driven Insight
The paper provides empirical evidence that the orthogonal component of the flatness gradient \(g_{vf}\) changes slowly: \[ g_{vf} = g_f \sin(\phi_f), \quad \text{with small temporal variation over iterations.} \] This observation underlies the caching mechanism and is supported by training-curve analysis.

**Weaknesses:**

1. Lack of Theoretical Justification
The claimed invariance of \(g_{vf}\) and its safe reuse is empirically observed but not theoretically proven.
No convergence or approximation-error bound is provided for the caching mechanism or the adaptive trigger.
In contrast, the original C-Flat offered a formal connection between \(R^{(1)}\rho(\theta)\) and the largest eigenvalue of the Hessian \(\lambda{\max}(H)\): \[ R^{(1)}\rho(\theta^\) = \rho^2 \lambda*{\max}(H), \] whereas C-Flat Turbo does not extend this analysis to its modified update rule.

2. Heuristic Nature of Performance Gains
The improvement in accuracy is attributed to “horizontal and vertical components of the oracle gradient” and “progressive update of sharpness gradients,” which are qualitative heuristics rather than rigorously derived results.

3. Limited Statistical Reliability
All experiments use a single fixed seed (1993) for class-order shuffling, without reporting variance or confidence intervals.
This makes it difficult to assess the statistical robustness of the observed accuracy differences (\(\approx 1{-}3\%\)).

4. No Formal Analysis of Reuse Error
The paper does not analyze how long the cached \(g_{vf}\) remains valid before its direction drifts, nor how this affects convergence.
Without a formal bound on the approximation error of reused gradients, stability guarantees remain absent.

5. Empirical-Only Validation
All justifications for the “better performance” rest on visualization (loss surface flattening) and ablations, with no analytical or theoretical explanation for why the heuristic accelerations lead to higher accuracy rather than degradation.

6. Comparative Scope
While the method is evaluated across several baselines, the dataset and model diversity (4 datasets, 2 architectures) is moderate compared to recent CL papers reporting 8–11 datasets and broader ablations.

**Questions:**

1. On Theoretical Justification of Gradient Reuse

The paper’s main efficiency gain comes from reusing the first-order flatness gradient’s orthogonal component \( g_{vf} \) across several iterations.

Could you elaborate on whether there exists — or could be derived — a formal bound on the deviation between cached and true gradients over time?

In particular, how do you ensure that the accumulation of approximation error from reused \( g_{vf} \) does not destabilize convergence in longer continual learning sequences?

2. On Empirical versus Theoretical Balance

The empirical results show consistent accuracy improvements despite the heuristic modifications to C-Flat.

Do you attribute this performance gain primarily to reduced noise in optimization (e.g., smoother gradient trajectories) or to a fundamentally different convergence behavior induced by the caching mechanism?

If the latter, can you provide theoretical or experimental evidence that the modified update dynamics reach flatter minima than C-Flat’s full recomputation?

3. On Reproducibility and Statistical Robustness

The experiments were conducted using a single random seed (1993) for class-order shuffling.

Could you clarify whether multi-seed or multi-run evaluations were attempted during development, and if so, how consistent were the performance trends?

Given that continual learning tasks are sensitive to seed variation, do you expect the observed improvements (≈1–3%) to hold under different random task orders or stochastic initialization conditions?

---

### Official Review · Reviewer_CSa3 · 2025-11-03

**Soundness:** 3
**Presentation:** 3
**Contribution:** 3
**Rating:** 6
**Confidence:** 4

**Summary:**

The paper develops an optimizer C-Flat Turbo, which improves upon a previously developed one named C-Flat.  This class of optimizers target continual learning, where the goal is to train a network on a stream of tasks, ensuring the learning of new tasks while avoiding catastrophic forgetting on old ones. This class of optimizers achieve it by directing model weights towards regions of "flat minima". In C-Flat, this necessitates backpropagating w.r.t. multiple different weights, costing 4x computational overhead of vanilla GD due to introducing two additional loss terms -- zeroth-order sharpness (change in loss when weights are perturbed), and first-order flatness (gradient magnitude when weights are perturbed). The latter itself requires computing two separate gradients $g_0$ and $g_1$, the former requires 1 ($g_s$), in addition to standard SGD gradient $g$. In their new proposed optimizer, the authors improves upon the efficiency of naive C-Flat. This is achieved via (1) caching a component of the gradient $g_1$ to be reused for $k$ steps, where $k$ is determined by a initial hyperparameter $k_0$ and gradually increases as number of tasks increase, and (2) adaptively triggering the computation of $g_1$ based on deviations of the $g_0$ from its EMA. A similar process is also done for $g_s$, based on previous works LookSAM and AE-SAM respectively. Combined, this results in an optimizer that is "about 2x the speed of C-Flat, and 0.6x that of SGD" (L377), and experiments show that C-Flat Turbo can achieve slightly better accuracy than C-Flat.

**Strengths:**

- The authors conducted in-depth empirical analysis and visualizations to motivate the proposed method, which greatly adds to the clarity and presentation of the method. In particular, each component of the method is well-motivated by empirical analysis, and parallels to prior work in sharpness-aware optimization (SAM) are clearly drawn to provide better intuition for why they are used for flatness gradients.

- Comprehensive experiments across multiple datasets, and with multiple CL methods, with clear improvement over the latency of C-Flat while not only maintaining, but even slightly improving, accuracy scores.

**Weaknesses:**

- Overall improvement of CL optimizers do not seem very significant, as opposed to improvements from apply different CL methods. For instance, comparing within "Typical" and "PTM-based" baseline (no C-Flat) methods in Table 1, we see much larger variations among different method. E.g. MEMO is almost 2x faster in Img/s compared to iCaRL while achieving large improvements across all datasets (e.g. almost 5 points on CIFAR100). In contrast, applying C-Flat Turbo often results in much smaller gains across all datasets, while slowing training by 2x). Despite latency optimizations from C-Flat, this seems like a hefty training time trade-off for a proportionately much smaller improvement.

- Scheduler requires knowing the number of tasks beforehand (knowing $N$), which is often not the case in real-world continual learning scenarios.

- Usage of math notations need to be more consistent, especially in equations (1)-(5) -- the paper commonly alternates between usage of $g$ and  $\nabla {\mathcal L}(\theta)$, $g_s$ and $\nabla \mathcal{L}(\theta + \epsilon_0^\ast)$, etc.

- Some of the gains in latency are actually based on LookSAM and AE-SAM, rather than from this work. From what I can tell, the novel components proposed here can only improve latency for computing $g_1$, which means speedups isolated to methods proposed here are upper-bounded by 1.33x (4 backprops / step -> 3+ backprops / step).

- More ablations of C-Flat Turbo is important for assessing which component of C-Flat Turbo is contributing most towards the accuracy) improvements -- is it from the changes to computation of the sharpness gradients, or that of the flatness gradients? I.e. there should be some additional rows between +C-Flat and +C-Flat Turbo in Table 3, e.g. (+C-Flat + LookSAM, C-Flat Turbo - LookSAM, etc.)

- Overall I do believe that this paper is valuable in offering useful empirical insights for reducing the computation time of C-Flat, but I feel that its contributions are limited by the relatively weak performance of C-Flat itself.

**Questions:**

Since C-Flat Turbo is an optimized version of C-Flat using a series of approximations for caching / avoiding gradient computations, do the authors have an explanation for why C-Flat Turbo here can achieve better results than the actual optimizer that it is attempting to approximate?

- Minor: Several usages of "... 1x speedup ..." in the paper, this is confusing, especially given that the paper also uses "... 2x speedup" in other areas.

---

### Note · Authors · 2025-11-14

I have read and agree with the venue's withdrawal policy on behalf of myself and my co-authors.